

# Consistency analysis of consciousness index and bispectral index in monitoring the depth of sevoflurane anesthesia in laparoscopic surgery

Miao Huo[1], Qian Zhang[2], Xingxing Zheng[1], Hui Wang[1], Ning Bai[1], Ruifen Xu[1] and Ziyu Zhao[1]

[1] Department of Anesthesiology, Shaanxi Provincial People's Hospital, Xi'an, China
[2] Department of Burn and Plastic Surgery, Shaanxi Provincial People's Hospital, Xi'an, China

## ABSTRACT

**Background:** The Index of Consciousness (IoC) is a new monitoring index of anesthesia depth reflecting the state of consciousness of the brain independently developed by China. The research on monitoring the depth of anesthesia mainly focuses on propofol, and bispectral index (BIS) is a sensitive and accurate objective index to evaluate the state of consciousness at home and abroad. This study mainly analyzed the effect of IoC on monitoring the depth of sevoflurane anesthesia and the consistency and accuracy with BIS when monitoring sevoflurane maintenance anesthesia.

**Objective:** To investigate the monitoring value of the Index of Consciousness (IoC) for the depth of sevoflurane anesthesia in laparoscopic surgery.

**Methods:** The study population consisted of 108 patients who experienced elective whole-body anesthesia procedures within the timeframe of April 2020 to June 2023 at our hospital. Throughout the anesthesia process, which encompassed induction and maintenance using inhaled sevoflurane, all patients were diligently monitored for both the Bispectral Index (BIS) and the Index of Consciousness (IoC). We conducted an analysis to assess the correlation between IoC and BIS throughout the anesthesia induction process and from the maintenance phase to the regaining of consciousness. To evaluate the predictive accuracy of IoC and BIS for the onset of unconsciousness during induction and the return of consciousness during emergence, we employed receiver operating characteristic (ROC) curve analysis.

**Results:** The mean difference between BIS and IoC, spanning from the pre-anesthesia induction phase to the completion of propofol induction, was 1.3 (95% Limits of Agreement [−53.4 to 56.0]). Similarly, during the interval from the initiation of sevoflurane inhalation to the point of consciousness restoration, the average difference between BIS and IoC was 0.3 (95% LOA [−10.8 to 11.4]). No statistically significant disparities were observed in the data acquired from the two measurement methodologies during both the anesthesia induction process and the journey from maintenance to the regaining of consciousness ($P > 0.05$). The outcomes of the ROC curve analysis disclosed that the areas under the curve (AUC) for prognosticating the occurrence of loss of consciousness were 0.967 (95% CI [0.935–0.999]) for BIS and 0.959 (95% CI [0.924–0.993]) for IoC, with optimal threshold values set at 81 (sensitivity: 88.10%, specificity: 92.16%) and 77

Corresponding author
Ziyu Zhao, zhaoziyu3362@163.com

(sensitivity: 79.55%, specificity: 95.45%) correspondingly. For the prediction of recovery of consciousness, the AUCs were 0.995 (95% CI [0.987–1.000]) for BIS and 0.963 (95% CI [0.916–1.000]) for IoC, each associated with optimal cutoff values of 76 (sensitivity: 92.86%, specificity: 100.00%) and 72 (sensitivity: 86.36%, specificity: 100.00%) respectively.

**Conclusion:** The monitoring of sevoflurane anesthesia maintenance using IoC demonstrates a level of comparability to BIS, and its alignment with BIS during the maintenance phase of sevoflurane anesthesia is robust. IoC displays promising potential for effectively monitoring the depth of anesthesia.

## INTRODUCTION

General anesthesia is a commonly used method for laparoscopic surgery. Nonetheless, when a patient's intraoperative Bispectral Index (BIS) drops below 40 and persists at this level for more than 5 min, their susceptibility to myocardial infarction, stroke, and mortality elevates (*Leslie et al., 2010*). Hence, appropriate anesthesia depth holds significant importance for patient prognosis. Previous studies have indicated (*Zhang et al., 2022*) that inadequate depth of anesthesia leads to intraoperative movement, affecting surgical procedures and increasing patient stress responses. Conversely, excessive anesthesia depth results in higher drug dosages, respiratory and circulatory depression, delayed recovery, and potentially increased postoperative complications, adversely impacting recovery. Traditional methods to monitor anesthesia depth predominantly hinge on factors like variations in hemodynamics and skeletal muscle reactions to regulate the administration rate of general anesthesia agents. Nevertheless, these indicators manifest considerable differences among individuals and are susceptible to effects stemming from vasopressor agents, blood volume status, preoperative fasting duration, and concurrent medications (*Chen et al., 2021*). Consequently, they lack precision in reflecting anesthesia depth and guiding clinical anesthesia management. Presently, various electroencephalographic monitoring techniques are employed in clinical practice, including BIS, Auditory Evoked Potential Index (AAI), and the Index of Consciousness (IoC). Numerous studies have demonstrated a strong correlation between these indices and anesthesia depth (*Cotae et al., 2021*; *Arnold et al., 2019*), offering distinct advantages over traditional methods that rely on vital signs for assessment. Among them, BIS is a sensitive and accurate objective index to evaluate the state of consciousness at home and abroad (*Shetty et al., 2018*), and IoC is a new monitoring index of anesthesia depth reflecting the state of consciousness of the brain independently developed by our country. It is a dimensionless value of 0~99 calculated by three EEG parameters based on the sample entropy of EEG information in complexity analysis, the edge frequency in frequency domain analysis and the burst suppression ratio in time domain analysis. It is recommended that the optimal interval of IoC is the same as that of BIS, which is 40–60, indicating that the depth of sedation is appropriate. *Qi et al. (2023)* pointed out that the

application of IoC monitoring in elderly patients undergoing laparoscopic urological surgery can reduce intraoperative stress, reduce peripheral and central inflammatory damage, and accelerate postoperative recovery. However, the research on IoC for monitoring the depth of anesthesia mainly focuses on propofol, and there are few reports on the effect of monitoring the depth of sevoflurane anesthesia and the consistency and accuracy with BIS when monitoring sevoflurane to maintain anesthesia. This study primarily focuses on exploring the monitoring utility of IoC in evaluating the extent of sevoflurane anesthesia within the context of laparoscopic surgery. The findings offer significant perspectives for its practical implementation in clinical settings.

## MATERIALS AND METHODS

### Study population

A total of 108 patients who underwent elective laparoscopic hysterectomy or cholecystectomy under general anesthesia at our institution between April 2020 and June 2023 were enrolled as study subjects. (1) Inclusion criteria: The age is between 18 and 65 years old; elective laparoscopic hysterectomy or cholecystectomy under general anesthesia; American Society of Anesthesiologists (ASA) Physical Status I–II; body mass index (BMI) was 18–39 kg/m$^2$; estimated surgery duration <6 h. (2) Exclusion criteria: Patients with allergies to anesthesia drugs; those with severe heart, lung, or kidney impairment; those with severe cardiovascular or cerebrovascular diseases; individuals with mental disorders; individuals on long-term sedative or hypnotic medication; preoperative history of nervous system. (3) Exclusion criteria: Patients who converted to open surgery during laparoscopic procedures; lost to follow-up. All samples obtained in this study were approved by the ethics committee of the Shaanxi Provincial People's Hospital and abided by the ethical guidelines of the Declaration of Helsinki, and ethics committee agreed to waive informed consent.

### Methods

#### Preoperative anesthetic preparation

All patients were routinely prohibited from oral intake for 2 h and fasting for 8 hours prior to surgery. Upon admission to the operating room, a peripheral intravenous line was established, and standard vital signs including heart rate (HR), electrocardiogram (ECG), blood pressure (BP), oxygen saturation (SpO$_2$), and mean arterial pressure (MAP) were monitored. Oxygen supplementation was provided *via* a facemask. Simultaneously, a domestically developed anesthesia depth monitoring device (Angel-6000D) and a BIS monitor were utilized in parallel to monitor IoC and BIS, respectively. The procedure involved placing BIS electrodes or IoC electrodes on the left or right side of the patient's forehead, and prior to attachment, the patient's forehead skin was cleansed with saline solution to minimize impedance.

#### Anesthesia procedure

Anesthetic induction: Anesthesia induction involved a stepwise intravenous administration of propofol at a dosage range of 1.5 to 3.0 mg/kg, sufentanil at a dose

of 0.1 to 0.2 μg/kg, and rocuronium bromide at a dosage of 0.6 to 0.9 mg/kg. Once muscle relaxation was achieved, facilitating endotracheal intubation, the anesthesia machine was connected, and respiratory control was initiated. Oxygen was supplied at a rate of 2 L/min with a tidal volume of 8 to 10 mL/kg, a respiratory rate set between 12 and 14 breaths per minute, and an inspiratory-expiratory ratio maintained at 1:2. Respiratory parameters were meticulously adjusted during the procedure to sustain PETCO2 levels between 30 to 40 mmHg.

Anesthetic maintenance: A continuous inhalation of sevoflurane at concentrations of 2% to 3% was consistently administered throughout the surgical procedure. Concurrently, a continuous infusion of remifentanil was maintained at a rate ranging from 0.1 to 0.2 μg/kg/min.

The BIS index was meticulously maintained within the range of 40 to 60. In instances where the patient's intraoperative heart rate (HR) dropped below 50 beats per minute, atropine was administered at a dose of 0.3 to 0.5 mg. Conversely, if the HR exceeded 100 beats per minute, esmolol was administered at a dose of 0.5 mg/kg, with additional dosing applied as required. Approximately 30 min before the conclusion of surgery, an intravenous administration of sufentanil at a dose of 0.1 to 0.5 μg/kg was administered.

Surgical conclusion and postoperative phase: At the completion of the surgical procedure, the administration of remifentanil and sevoflurane was promptly terminated. Following extubation, a regimen of intravenous patient-controlled analgesia (PCIA) was initiated for all patients. This entailed a mixture of sufentanil (100 μg), flurbiprofen ester (200 mg), and ondansetron hydrochloride (32 mg) in a total volume of 100 mL. The initial infusion rate was set at 2 mL/h, with a patient-controlled rate of 1.5 mL/h and a maximum patient-controlled dose of 8 mL/h, along with a lockout time of 20 min.

Subsequent to surgery, patients were transitioned to the Post-Anesthesia Care Unit (PACU) for vigilant recovery monitoring.

## Observation indices

(1) General information:

Basic demographic data, such as gender, age, body mass index (BMI), American Society of Anesthesiologists (ASA) classification, and surgical duration, will be collected as part of the general patient information.

(2) Recording of BIS and IoC at specific time points:

The monitoring of anesthesia depth will occur at specific time intervals, encompassing critical stages of the surgical procedure. These time points include:

Before the initiation of anesthetic induction (T0).

At the point of loss of consciousness (T1).

Before intubation (T2).

During intubation (T3).

Throughout sevoflurane inhalation (T4).

At the moment of skin incision (T5).

Upon the completion of abdominal insufflation (T6).

A total of 1 min after the commencement of abdominal irrigation (T6).

Concluding abdominal insufflation (T7).

At the conclusion of surgery (T8).

When consciousness is regained during emergence from anesthesia (T9).

The concordance between the IoC and BIS readings during these specified time points will be meticulously analyzed.

(3) Receiver operating characteristic (ROC) curve analysis for IoC and BIS:

ROC curve analysis will be conducted to determine the predictive capabilities of both IoC and BIS regarding the occurrence of loss of consciousness during anesthetic induction and the subsequent recovery of consciousness during the emergence from anesthesia. Loss of consciousness during anesthetic induction will be defined as the absence of response to verbal commands, while recovery of consciousness will be defined as the resumption of response to verbal commands.

(4) Assessment of intraoperative awareness using the Modified Brice Questionnaire:

In order to assess the occurrence of intraoperative awareness, patients will be queried on the first and third postoperative days using the Modified Brice Questionnaire. This questionnaire comprises five questions and patients' responses will be categorized as "yes," "no," or "suspected" based on their answers.

## Statistical analysis

Statistical analysis was conducted using SPSS version 22.0 software. Continuous variables were presented as means ± standard deviations. Parametric data were compared using the t-test. Categorical variables were expressed as either counts or percentages and were analyzed using the chi-square test. In cases where the theoretical frequency was less than or equal to 5 but greater than or equal to 1, the chi-square value was adjusted. When the theoretical frequency was less than 1, Fisher's exact test was applied. To assess the concordance between the two anesthesia depth indices, the Bland-Altman method was employed. The mean difference between the two indices as well as the 95% limits of agreement (LOA) were calculated. ROC curve analysis was utilized to evaluate the predictive value of both BIS and IoC concerning patients' consciousness status. A significance level of $P < 0.05$ was considered indicative of statistically significant differences.

## RESULTS

### Patient demographics

A total of 108 patients were ultimately included in this study, comprising 19 males and 89 females. The patients' ages ranged from 39 to 68 years, with a mean age of $(51.27 \pm 6.84)$ years. The body mass index (BMI) was $(24.58 \pm 1.63)$ kg/m$^2$. ASA classifications were as follows: Class I (20 cases) and Class II (88 cases). The surgical procedures consisted of 84 cases of laparoscopic hysterectomy and 24 cases of laparoscopic cholecystectomy. The mean surgical duration was $121.53 \pm 41.36$ min.

**Table 1  Analysis of consistency between IoC and BIS during anesthetic induction (x ± s).**

| Index | n | T0 | T1 | T2 | T3 |
|---|---|---|---|---|---|
| IoC | 108 | 89.91 ± 2.56 | 61.55 ± 5.89 | 51.63 ± 4.15 | 50.08 ± 3.63 |
| BIS | 108 | 90.24 ± 2.74 | 60.34 ± 4.82 | 52.78 ± 3.86 | 51.44 ± 2.48 |
| t | | 0.915 | 1.652 | 2.109 | 3.215 |
| P | | 0.361 | 0.100 | 0.036 | 0.002 |

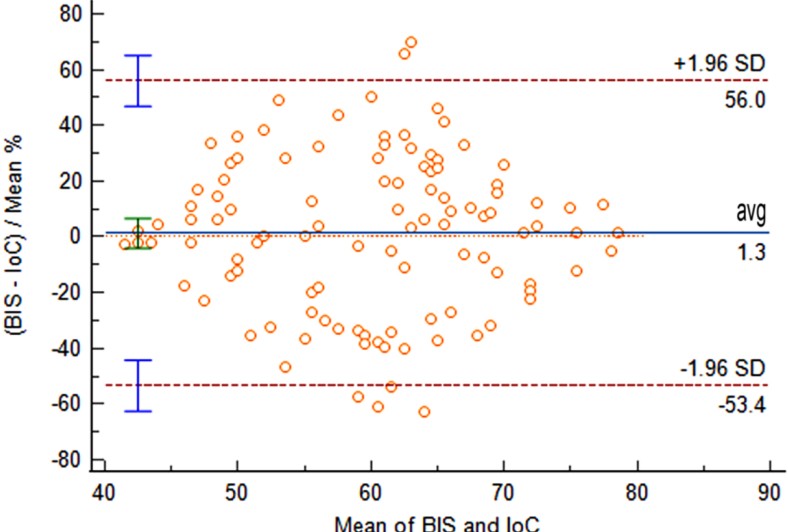

**Figure 1  Bland-Altman analysis of consistency between IoC and BIS during anesthetic induction.**

## Analysis of consistency between IoC and BIS during anesthetic induction

Comparison of IoC and BIS values at time points T0 and T1 revealed no statistically significant differences ($P > 0.05$). At time points T2 and T3, IoC values were significantly lower than BIS values, with statistically significant differences ($P < 0.05$), as shown in Table 1. From the pre-anesthetic induction (T0) to the completion of propofol induction, the mean difference between BIS and IoC was 1.3 (95% LOA [−53.4 to 56.0]). The comparison of data obtained from the two measurement methods yielded non-significant differences ($P = 0.628$), indicating good consistency between IoC and BIS during the anesthetic induction process, as illustrated in Fig. 1.

## Analysis of consistency between IoC and BIS during anesthetic maintenance to recovery of consciousness stage

At time points T4 to T9, IoC values were consistently lower than BIS values, with statistically significant differences ($P < 0.05$), as shown in Table 2. From the initiation of sevoflurane inhalation to the recovery of consciousness, the mean difference between BIS and IoC was 0.3 (95% LOA [−10.8 to 11.4]). The comparison of data obtained from the two

**Table 2 Analysis of consistency between IoC and BIS during anesthetic maintenance to recovery of consciousness stage (x ± s).**

| Index | n | T4 | T5 | T6 | T7 | T8 | T9 |
|---|---|---|---|---|---|---|---|
| IoC | 108 | 54.89 ± 2.36 | 53.29 ± 2.44 | 51.68 ± 2.81 | 52.38 ± 2.57 | 52.45 ± 2.72 | 51.88 ± 2.87 |
| BIS | 108 | 55.83 ± 2.48 | 54.61 ± 2.97 | 52.89 ± 3.12 | 53.67 ± 2.45 | 53.82 ± 2.23 | 53.64 ± 2.38 |
| t | | 2.853 | 3.569 | 2.995 | 3.776 | 4.048 | 4.906 |
| P | | 0.005 | <0.001 | 0.003 | <0.001 | <0.001 | <0.001 |

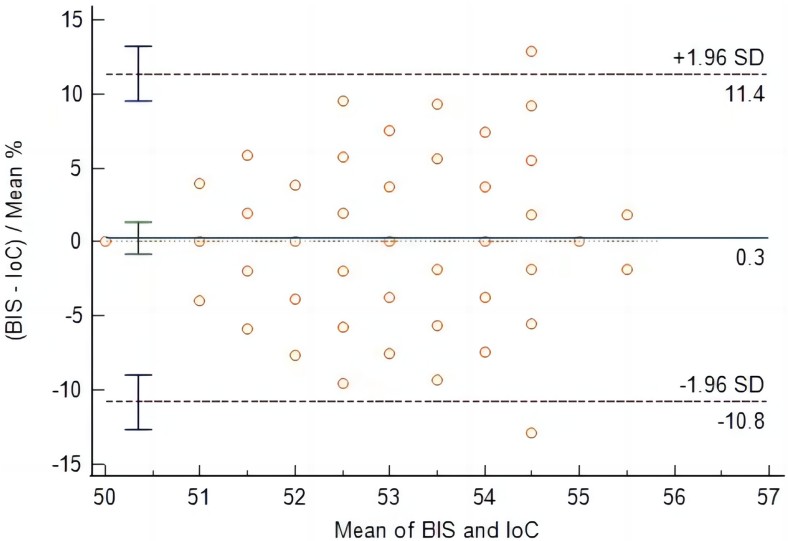

**Figure 2 Bland-Altman analysis of consistency between IoC and BIS during anesthetic maintenance to recovery of consciousness stage.**

measurement methods yielded non-significant differences ($P = 0.604$), indicating good consistency between IoC and BIS during the anesthetic maintenance to recovery of consciousness stage, as illustrated in Fig. 2.

## ROC curve analysis of BIS and IoC for predicting loss of consciousness

The results of the ROC curve analysis revealed that the area under the curve (AUC) for BIS and IoC in monitoring loss of consciousness were 0.967 (95% CI [0.935–0.999]) and 0.959 (95% CI [0.924–0.993]), respectively. The optimal cutoff values were determined to be 81 (sensitivity: 88.10%, specificity: 92.16%) for BIS and 77 (sensitivity: 79.55%, specificity: 95.45%) for IoC, as depicted in Fig. 3.

## Bispectral index and index of consciousness for predicting patient consciousness recovery: ROC curve analysis

The ROC curve analysis results revealed that the areas under the curve (AUC) for monitoring consciousness recovery using BIS and IoC were 0.995 (95% CI [0.987–1.000]) and 0.963 (95% CI [0.916–1.000]), respectively. The optimal threshold values were

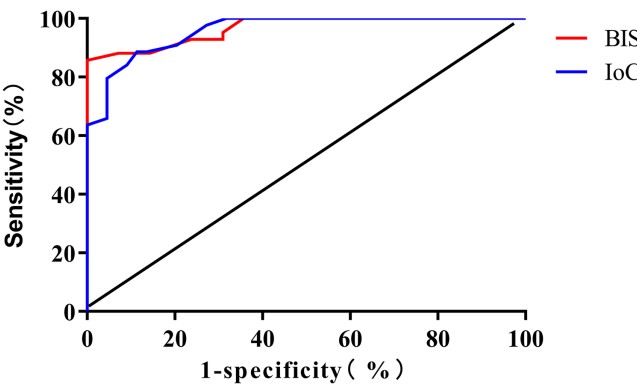

**Figure 3** ROC curve analysis for predicting patient loss of consciousness using BIS and IoC.

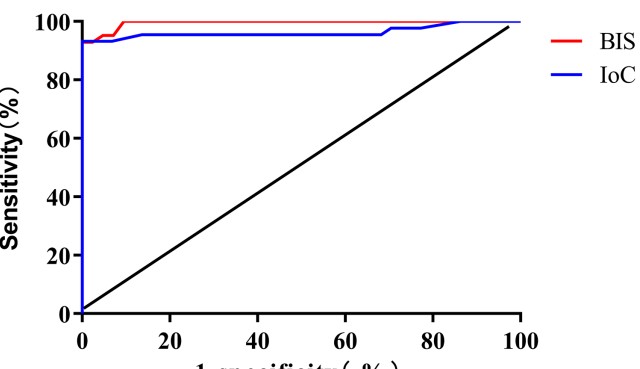

**Figure 4** ROC curve analysis for predicting patient consciousness recovery using BIS and IoC.

determined to be 76 (sensitivity: 92.86%, specificity: 100.00%) for BIS and 72 (sensitivity: 86.36%, specificity: 100.00%) for IoC (Fig. 4).

### Intraoperative awareness

No intraoperative awareness and other anesthesia complications occurred in all patients.

## DISCUSSION

General anesthesia refers to a pathophysiological state where anesthetic agents are introduced into the patient's body *via* inhalation, intravenous injection, or intramuscular injection, resulting in central nervous system suppression, disappearance of consciousness, and absence of pain sensation. An ideal general anesthesia should achieve objectives such as complete loss of patient consciousness, effective pain control, moderate muscle relaxation, appropriate control of stress responses, and relatively stable internal environment to meet surgical requirements and ensure patient safety (*Wijnberge et al., 2020*). Depth of anesthesia monitoring has become a focal point in current anesthesia research. Effective depth of anesthesia monitoring aids in stabilizing intraoperative hemodynamics, reducing anesthesia drug dosages, shortening postoperative recovery time,

and decreasing the incidence of intraoperative adverse reactions and postoperative complications (*Xuan & Xu, 2022*). The intricate structure and functions of the brain present a challenge in depth of anesthesia monitoring. Presently, various brain electrical monitoring techniques are utilized in clinical settings, including Bispectral Index (BIS), Narcotrend Index (NI), Auditory Evoked Potentials Index (AAI), Index of Consciousness (IoC), among others. While there is no absolute gold standard, numerous studies suggest that these indices exhibit strong correlations with anesthesia depth, showing superiority over traditional methods based on vital signs for depth assessment (*Li et al., 2020*; *Liu et al., 2023*).

BIS is currently the sole anesthesia sedation depth monitoring parameter that has received approval from the U.S. Food and Drug Administration (FDA). Currently, BIS monitoring for anesthesia depth has been widely adopted and serves as a routine means of clinical anesthesia depth monitoring. Extensive research has indicated (*Lewis et al., 2019*; *Romito et al., 2022*) that BIS effectively monitors the functional state and alterations of the cerebral cortex, demonstrating a certain level of sensitivity in predicting body movements, intraoperative awareness, as well as the onset and recovery of consciousness. Simultaneously, it has the potential to reduce the dosage of anesthesia drugs, shorten postoperative recovery and extubation times. IoC, a novel anesthesia depth monitoring parameter in China, accurately reflects the depth of propofol anesthesia. Its equipment is user-friendly, operationally straightforward, requires no consumables, and is relatively cost-effective (*Jensen et al., 2008*). It is recommended that the optimal interval of IoC is the same as that of BIS, which is 40–60, indicating that the depth of sedation is appropriate.

In this study, patients undergoing laparoscopic cholecystectomy or uterine appendectomy were selected as subjects. His selection was primarily guided by the benefits associated with these procedures, including minimal trauma, reduced bleeding, stable intraoperative hemodynamics, relatively short operation times, and swift postoperative recovery (*Liang et al., 2021*). Moreover, the synchronized monitoring of IoC and BIS within the same patients was employed to eliminate inter-individual variability in drug responses, enabling the simultaneous observation of IoC's trends and accuracy relative to BIS. By analyzing the consistency between IoC and BIS for sevoflurane anesthesia depth during laparoscopic surgery, the results indicated that from pre-induction to completion of propofol anesthesia induction, and from sevoflurane inhalation to recovery of consciousness, the data obtained from BIS and IoC comparisons exhibited no significant differences ($P > 0.05$), demonstrating strong consistency. Both indices could effectively guide intraoperative sevoflurane concentration adjustments to prevent excessive or insufficient anesthesia. The study results revealed that the optimal threshold values for monitoring loss of consciousness were 81 (sensitivity: 88.10%, specificity: 92.16%) for BIS and 77 (sensitivity: 79.55%, specificity: 95.45%) for IoC; while for monitoring recovery of consciousness, the optimal threshold values were 76 (sensitivity: 92.86%, specificity: 100.00%) for BIS and 72 (sensitivity: 86.36%, specificity: 100.00%) for IoC, indicating that IoC's monitoring of sevoflurane anesthesia maintenance depth was equivalent to BIS.

The study observed that the IoC values were consistently lower than BIS values at time points T2 to T9, with statistically significant differences ($P < 0.05$). This could be attributed

to differences in the sensitivity of BIS and IoC to opioid drugs (*Wang et al., 2020*).It has been reported that (*Zhan et al., 2023*), high-dose remifentanil has no obvious effect on BIS, while remifentanil can reduce IoC, which may be the reason why IoC value is significantly lower than BIS value at T2~T9. The results of *Jensen et al. (2014)* showed that in total intravenous anesthesia with propofol and remifentanil, IoC can predict the moment when the patient's consciousness disappears (the moment when the patient's eyelash reflex disappears), which is similar to the results of this study, indicating that IoC has a good value in predicting the disappearance of surgical consciousness.

The present study does exhibit certain limitations. For instance, the monitoring of muscle relaxation status was not undertaken, and it should be acknowledged that muscle relaxation can indeed influence anesthesia depth monitoring. Additionally, the choice of surgical procedures in this research encompassed laparoscopic cholecystectomy or uterine appendectomy, which may not comprehensively represent all laparoscopic surgeries. This selection potentially restricts the broader application of IoC for monitoring anesthesia depth.

# CONCLUSIONS

The depth of anesthesia monitored by IoC during the maintenance of sevoflurane anesthesia is comparable to that of BIS, and has a good consistency with the depth of anesthesia monitored by BIS during the maintenance of sevoflurane anesthesia. The application prospect of IoC monitoring the depth of anesthesia is good, which can better reflect the depth of anesthesia in patients undergoing laparoscopic surgery. It is of great value for guiding clinical anesthesia, preventing intraoperative awareness and accelerating postoperative recovery.

## Funding
The authors received no funding for this work.

## Competing Interests
The authors declare that they have no competing interests.

## Author Contributions
- Miao Huo conceived and designed the experiments, performed the experiments, analyzed the data, prepared figures and/or tables, authored or reviewed drafts of the article, and approved the final draft.
- Qian Zhang performed the experiments, analyzed the data, prepared figures and/or tables, and approved the final draft.
- Xingxing Zheng conceived and designed the experiments, authored or reviewed drafts of the article, and approved the final draft.
- Hui Wang performed the experiments, authored or reviewed drafts of the article, and approved the final draft.

**Lewis SR, Pritchard MW, Fawcett LJ, Punjasawadwong Y. 2019.** Bispectral index for improving intraoperative awareness and early postoperative recovery in adults. *The Cochrane Database of Systematic Reviews* **9(9)**:CD003843 DOI 10.1002/14651858.CD003843.pub4.

**Li R, Wu Q, Liu J, Wu Q, Li C, Zhao Q. 2020.** Monitoring depth of anesthesia based on hybrid features and recurrent neural network. *Frontiers in Neuroscience* **14**:26 DOI 10.3389/fnins.2020.00026.

**Liang T, Wu F, Wang B, Mu F. 2021.** PRISMA: accuracy of response entropy and bispectral index to predict the transition of consciousness during sevoflurane anesthesia: a systematic review and meta-analysis. *Medicine* **100(17)**:e25718 DOI 10.1097/MD.0000000000025718.

**Liu Z, Si L, Li J, Zhu J, Lee WH, Chen B, Yan X, Wang Q, Wang G. 2023.** Monitoring the depth of anesthesia based on phase-amplitude coupling of near-infrared spectroscopy signals. *IEEE Transactions on Neural Systems and Rehabilitation Engineering: A Publication of the IEEE Engineering in Medicine and Biology Society* **31**:2849–2857 DOI 10.1109/TNSRE.2023.3289183.

**Qi F, Fan L, Wang C, Liu Y, Yang S, Fan Z, Miao F, Kan M, Feng K, Wang T. 2023.** Index of consciousness monitoring during general anesthesia may effectively enhance rehabilitation in elderly patients undergoing laparoscopic urological surgery: a randomized controlled clinical trial. *BMC Anesthesiology* **23(1)**:331 DOI 10.1186/s12871-023-02300-z.

**Romito JW, Atem FD, Manjunath A, Yang A, Romito BT, Stutzman SE, McDonagh DL, Venkatachalam AM, Premachandra L, Aiyagari V. 2022.** Comparison of bispectral index monitor data between standard frontal-temporal position and alternative nasal dorsum position in the intensive care unit: a pilot study. *The Journal of Neuroscience Nursing: Journal of the American Association of Neuroscience Nurses* **54(1)**:30–34 DOI 10.1097/JNN.0000000000000635.

**Shetty RM, Bellini A, Wijayatilake DS, Hamilton MA, Jain R, Karanth S, Namachivayam A. 2018.** BIS monitoring versus clinical assessment for sedation in mechanically ventilated adults in the intensive care unit and its impact on clinical outcomes and resource utilization. *The Cochrane Database of Systematic Reviews* **2(2)**:CD011240 DOI 10.1002/14651858.CD011240.pub2.

**Wang X, Zhang J, Feng K, Yang Y, Qi W, Martinez-Vazquez P, Zhao G, Wang T. 2020.** The effect of hypothermia during cardiopulmonary bypass on three electro-encephalographic indices assessing analgesia and hypnosis during anesthesia: consciousness index, nociception index, and bispectral index. *Perfusion* **35(2)**:154–162 DOI 10.1177/0267659119864821.

**Wijnberge M, Geerts BF, Hol L, Lemmers N, Mulder MP, Berge P, Schenk J, Terwindt LE, Hollmann MW, Vlaar AP, Veelo DP. 2020.** Effect of a machine learning-derived early warning system for intraoperative hypotension vs standard care on depth and duration of intraoperative hypotension during elective noncardiac surgery: the HYPE randomized clinical trial. *JAMA* **323(11)**:1052–1060 DOI 10.1001/jama.2020.0592.

**Xuan H, Xu K. 2022.** Warning and nursing experience of anesthesia depth monitoring for patients with general anesthesia delayed to leave anesthesia recovery room and delirium. *Emergency Medicine International* **2022(10)**:3610838 DOI 10.1155/2022/3610838.

**Zhan J, Chen F, Wu Z, Duan Z, Deng Q, Zeng J, Hou L, Zhang J, Si Y, Liu K, Wang M, Li H. 2023.** Consistency of the anesthesia consciousness index versus the bispectral index during laparoscopic gastrointestinal surgery with sevoflurane anesthesia: a prospective multi-center randomized controlled clinical study. *Frontiers in Aging Neuroscience* **15**:1084462 DOI 10.3389/fnagi.2023.1084462.

**Zhang X, Xu M, Li X, Cao X, Che X. 2022.** Application of intelligent detection of neural signal in depth evaluation of obstetrics and gynecology anesthesia. *Contrast Media & Molecular Imaging* **2022(1)**:6027965 DOI 10.1155/2022/6027965.