# Peer review of "Consistency analysis of consciousness index and bispectral index in monitoring the depth of sevoflurane anesthesia in laparoscopic surgery"

_PeerJ, doi:10.7717/peerj.16848_

## Round 0.1 · original submission · Major Revisions

Please revise the manuscript as the reviewers suggested.

Reviewer 1 ·

Basic reporting

The manuscript presents a detailed investigation into the monitoring value of the Consciousness Index (IoC) for the depth of sevoflurane anesthesia in laparoscopic surgery. The study included 108 patients undergoing elective whole-body anesthesia procedures, with a focus on the consistency and accuracy of IoC compared to Bispectral Index (BIS) during sevoflurane maintenance. The authors meticulously analyzed the correlation between IoC and BIS at various stages of anesthesia induction and maintenance, providing valuable insights into the potential of IoC in effectively monitoring the depth of anesthesia, particularly in the context of laparoscopic surgery. The manuscript is written in clear and technically correct English, offering a comprehensive introduction and background that contextualizes the study within the broader field of anesthesia monitoring. The study addresses a significant knowledge gap by specifically examining the applicability of IoC in monitoring sevoflurane anesthesia depth during laparoscopic surgery, filling a critical void in the existing literature. The methods employed are of a high technical standard, and the level of detail provided enhances the replicability of the study. However, the manuscript would benefit from greater detail in discussing the statistical analyses performed and in evaluating the robustness and control measures of the underlying data. Overall, the manuscript offers valuable contributions to the field of anesthesia depth monitoring and presents a well-executed study with significant implications for clinical practice.

Experimental design

1. The study population should be described in more detail in the inclusion and exclusion criteria section, including demographics and relevant medical history.
2. More detailed information on the anesthesia maintenance and recovery phase should be provided, particularly regarding the administration of anesthesia and the monitoring of BIS and IoC values

Validity of the findings

1. The abstract should briefly mention the key findings added to previous studies to give readers a quick overview of the results, conclusions and innovation.
2. The introduction should include a paragraph summarizing the current state of research on the topic and how this study fits into the existing knowledge.
3. The introduction should include detailed information on previous research on the Consciousness Index (IoC) and its application in anesthesia depth monitoring, specifically related to laparoscopic surgery.
4. The results section should clearly present the primary findings of the study, including the mean differences between BIS and IoC at various time points and the implications of these findings.

Additional comments

No comment.

Reviewer 2 ·

Basic reporting

So, I read this paper that's all about this new thing called the Consciousness Index (IoC) and how it's used to monitor the depth of sevoflurane anesthesia during laparoscopic surgery. The study fills a gap in the research by looking at how IoC compares to the Bispectral Index (BIS) in monitoring anesthesia depth during sevoflurane administration. The technical standard and method used in the study are pretty solid, making it easy for other researchers to replicate. The statistical analysis is sound, and they've got a good handle on controlling the data. Overall, the paper gives a really solid overview of IoC and its potential for monitoring anesthesia depth.

Experimental design

a) Please provide a rationale for choosing laparoscopic hysterectomy and cholecystectomy as the specific surgical procedures for this study.

Validity of the findings

a) The introduction provides a comprehensive overview of the importance of monitoring anesthesia depth in laparoscopic surgery and the limitations of traditional methods. However, it should also clearly outline the specific objectives of this study.
b) The discussion should compare the study's findings with previous literature on anesthesia depth monitoring and address any discrepancies or similarities.
c) Provide detailed information on any adverse events or complications related to anesthesia observed during the study.

Additional comments

a) Please discuss the potential clinical implications of the study's findings and how they could impact the practice of anesthesia in laparoscopic surgery.

Reviewer 3 ·

Basic reporting

This manuscript presents a study comparing the Consciousness Index (IoC) with the Bispectral Index (BIS) for monitoring the depth of sevoflurane anesthesia in laparoscopic surgery. The study aims to fill a knowledge gap by exploring the application of IoC in this context. The technical standard and method employed are robust, allowing for replicability. The statistical analysis is sound, and the control measures of the underlying data are well-verified. However, the study could benefit from a more comprehensive discussion of the limitations and potential biases. Additionally, further exploration of the broader implications and potential challenges in implementing IoC in clinical settings could enhance the manuscript's impact.

Experimental design

1) Clarify the significance of the mean surgical duration and its potential impact on the study's outcomes.
2) Address the limitations of the study in the discussion, including potential sources of bias and confounding factors.
3) Provide a rationale for the choice of laparoscopic hysterectomy and cholecystectomy as the specific surgical procedures for this study.

Validity of the findings

1) There is a lack of discussion about the normal range of BIS and IoC values and how the study values compare to these ranges. Besides, elaborate on the potential factors contributing to the differences in IoC and BIS values observed during the study, such as patient demographics, type of surgery, or concurrent medications.
2) Provide a clear conclusion that summarizes the key findings and their potential impact on anesthesia practice.
3) Consider discussing the potential implications of the study's findings on patient outcomes, such as postoperative recovery and complication rates.

Additional comments

1) The introduction could benefit from a more concise structure, with separate sections for background information and the specific aims of this study. Besides, the introduction contains a lot of background information on general anesthesia but lacks specific information about the study's focus on monitoring the depth of sevoflurane anesthesia in laparoscopic surgery.
2) Include recommendations for future research based on the limitations identified in the study.

---

## Round 0.2 · accepted · Accept

This manuscript can be accepted now.

Reviewer 1 ·

Basic reporting

I have reviewed the revised manuscript and found that the author has made detailed revisions to the basic report, experimental design, and validity sections of the research results, and has provided point-to-point responses to my comments. I have no further comments on this article, I believe it can be published.

Experimental design

No comment.

Validity of the findings

No comment.

Additional comments

No comment.

Reviewer 2 ·

Basic reporting

I have reviewed the revised manuscript and responded to the review comments very well. The author also provided explanations for these comments, and I believe there are no further issues with this article.

Experimental design

In terms of experimental design, the author made further modifications and achieved a good level.

Validity of the findings

Detailed supplements have also been made in terms of results.

Additional comments

I have no further review comments, the article can be published.

Reviewer 3 ·

Basic reporting

The article has a clear and concise structure, and professional English is used throughout the entire process.
The literature references provide sufficient on-site background.
The chart is clear and in compliance with regulations.

Experimental design

The research question is clearly defined, relevant, and meaningful. It illustrates how research can fill identified knowledge gaps.
The described method has sufficient details and information for replication.

Validity of the findings

All basic data has been provided; They are robust, statistically reliable, and controllable.
The conclusion statement is sufficient and relevant to the original research question, limited to supporting results.

Additional comments

No comment.